# The Role of Opioid Receptor Antagonists in Regulation of Blood Pressure and T-Cell Activation in Mice Selected for High Analgesia Induced by Swim Stress

**DOI:** 10.3390/ijms25052618

**Published:** 2024-02-23

**Authors:** Dominik Skiba, Kinga Jaskuła, Agata Nawrocka, Piotr Poznański, Marzena Łazarczyk, Łukasz Szymański, Tymoteusz Żera, Mariusz Sacharczuk, Agnieszka Cudnoch-Jędrzejewska, Zbigniew Gaciong

**Affiliations:** 1Department of Experimental Genomics, Institute of Genetics and Animal Biotechnology, Polish Academy of Sciences, Postepu 36A Street, Jastrzebiec, 05-552 Magdalenka, Poland; k.jaskula@igbzpan.pl (K.J.); a.nawrocka@igbzpan.pl (A.N.); p.poznanski@igbzpan.pl (P.P.); m.lazarczyk@igbzpan.pl (M.Ł.); l.szymanski@igbzpan.pl (Ł.S.); m.sacharczuk@igbzpan.pl (M.S.); 2Department of Experimental and Clinical Physiology, Center for Preclinical Research, Medical University of Warsaw, Banacha 1B Street, 02-097 Warsaw, Poland; tymoteusz.zera@wum.edu.pl (T.Ż.); agnieszka.cudnoch@wum.edu.pl (A.C.-J.); 3Department of Pharmacodynamics, Medical University of Warsaw, Zwirki i Wigury 81 Street, 02-091 Warsaw, Poland; 4Department and Clinic of Internal Diseases, Hypertension and Angiology, Medical University of Warsaw, Banacha 1A Street, 02-097 Warsaw, Poland; zbigniew.gaciong@wum.edu.pl

**Keywords:** blood pressure, opioids, naloxone, T-cell

## Abstract

Opioid peptides and their G protein-coupled receptors are important regulators within the cardiovascular system, implicated in the modulation of both heart and vascular functions. It is known that naloxone—an opioid antagonist—may exert a hypertensive effect. Recent experimental and clinical evidence supports the important role of inflammatory mechanisms in hypertension. Since opioids may play a role in the regulation of both blood pressure and immune response, we studied these two processes in our model. We aimed to evaluate the effect of selective and non-selective opioid receptor antagonists on blood pressure and T-cell activation in a mouse model of high swim stress-induced analgesia. Blood pressure was measured before and during the infusion of opioid receptor antagonists using a non-invasive tail–cuff measurement system. To assess the activation of T-cells, flow cytometry was used. We discovered that the non-selective antagonism of the opioid system by naloxone caused a significant elevation of blood pressure. The selective antagonism of μ and κ but not δ opioid receptors significantly increased systolic blood pressure. Subsequently, a brief characterization of T-cell subsets was performed. We found that the blockade of μ and δ receptors is associated with the increased expression of CD69 on CD4 T-cells. Moreover, we observed an increase in the central memory CD4 and central memory CD8 T-cell populations after the δ opioid receptor blockade. The antagonism of the μ opioid receptor increased the CD8 effector and central memory T-cell populations.

## 1. Introduction

The opioid system plays a particular role in hypertension development. Opioid peptides and their G protein-coupled receptors (GPCRs) are important regulators within the cardiovascular system, implicated in the modulation of electrophysiological function, heart rate, myocardial inotropy, and vascular function [1]. The activation of peripheral opioid receptors may mediate some of these actions, but others may result from direct or receptor-independent actions on cardiac tissue and the peripheral vascular system. There are three main types of opioid receptors—μ, κ, and δ—they can be further divided into several pharmacological subtypes. The most common approach to studying the opioid system is to completely block the whole opioid system, which is possible via the use of non-selective antagonists e.g., naloxone (NAL) or naltrexone. One of the first observed effects of NAL on the cardiovascular system was the elevation of blood pressure (BP), described nearly 50 years ago [2]. Over the following decades, more experimental work was published describing the hypertensive effect of NAL in various species, such as rat, dog, and cat, along with studies on humans [3,4,5,6,7]. However, some studies described a drop or no effect on BP levels after NAL infusion [8,9]. This inconsistency in results depends on the model used for the study, the dose and time of NAL infusion, the route of administration, etc. NAL is widely used in the treatment of severe opioid intoxication or the therapy of opioid-addicted patients [3]. Several studies have demonstrated a link between chronic pain and hypertension [4,5,6]. In both conditions, the opioid system plays an important role. Chronic pain is accompanied not only by an increase in BP but also by the development of chronic inflammation, which has been described recently as a novel factor in the development of hypertension [7]. Opioids may modulate the immune system, and NAL was found to be able to inhibit immune cell function [8]. Most of the research investigating the role of opioid system in the regulation of BP and immune system activation uses models with the administration of exogenous opioids. However, this tool is not effective for studying the role of endogenous opioids. Therefore, in our study, we used a mouse model with an enhanced endogenous opioid system—mice selected for high swim-stress-induced analgesia (HA) [9]. The model includes mice selected for over a hundred generations to obtain mice with a divergent opioid system activity: low and high. Due to using only pharmacological antagonists in this work, we did not use a mouse line with low opioid system activity, which has significantly affected G protein coupling to opioid receptors. HA mice are characterized by a higher expression of peripheral β-endorphin levels that could activate the central as well as the peripheral opioid systems [10]. Furthermore, those mice had a higher sensitivity to opioids, potentially due to a higher activity of G proteins in the brain [11,12]. In our previous study, we found that HA mice had an increased BP when exposed to NAL for one week. Interestingly, mice with low swim stress-induced analgesia (LA) had a higher basal BP level than HA mice, but NAL treatment had no effect on BP [13]. Therefore, in this study, we aimed to further investigate this effect using selective opioid receptor antagonists to evaluate which opioid receptors (μ, δ, or κ) influence the BP level. Moreover to elucidate the effect of peripheral opioid receptors blockade, we used naloxone methiodide (NAL-M). NAL-M is a non-selective opioid receptors antagonist, which does not cross blood-brain barrier (BBB). Additionally, taking into account the important role of the immune system in BP regulation and opioid analgesia, we aimed to evaluate the effect of selective opioid receptor antagonists on T-cell activation in this model.

## 2. Results

### 2.1. Effect of the Non-Selective Blockade of the Opioid System on Systolic and Diastolic Blood Pressure

Systolic BP and diastolic BP were measured over 10 consecutive days (Figure 1). Two-way RM-ANOVA revealed that systolic BP was stable during the whole experiment as indicated by non-significant time factor [F(1.89, 15.15) = 0.77; *p* = 0.47 for NAL and F(2.71, 21.67) = 1.61; *p* = 0.22 for NAL-M]. Additionally, RM-ANOVA within the control group (time as an independent factor) confirmed that systolic BP was stable across the whole experiment [F(2.36, 9.45) = 1.19; *p* = 0.35]. Non-selective antagonism of the opioid system by NAL [F(1, 8) = 12.01; *p* < 0.01], as well as NAL-M [F(1, 8) = 9.34; *p* < 0.05], had a significant effect on systolic BP but NAL effect was more prominent. In the case of both substances, the effect of the opioid system blockade remained stable during the whole experiment [F(9, 72) = 0.65; *p* = 0.75—time x NAL interaction and F(9, 72) = 1.43; *p* = 0.19—time x NAL-M].

Diastolic BP was also stable across the whole experiment, as shown by the non-significant time factor for NAL [F(2.32, 18.60) = 0.17; *p* = 0.87] and NAL-M [F(2.88, 23.05) = 1.77; *p* = 0.18]. Analysis within the control group (RM-ANOVA, time as an independent factor) proved that in control individuals, diastolic BP was stable during the whole experiment [F(2.89, 11.57) = 1.59; *p* = 0.25]. As with systolic BP, diastolic BP was elevated after non-selective antagonism of the opioid system [F(1, 8) = 7.57; *p* < 0.05—NAL and F(1, 8) = 15.55; *p* < 0.01—NAL-M] but, surprisingly, the effect of NAL-M was more prominent. The effect of pharmacotherapy was stable across time, as revealed by non-significant time × treatment interaction [F(9, 72) = 0.66; *p* = 0.74—NAL and F(9, 72) = 1.36; *p* = 0.22—NAL-M].

Additionally, mean systolic BP and diastolic BP for the whole period of treatment were calculated for each animal. In a further step, those values were compared by one-way ANOVA (treatment as a factor) (Appendix A). Mean systolic BP for the whole period of treatment was elevated after non-selective opioid system antagonism (CTRL: 102.72 mmHg ± 2.39 mmHg; NAL: 123.90 mmHg ± 5.63 mmHg; NAL-M: 120.79 mmHg ± 5.39 mmHg) as shown by significant treatment factor [F(2, 12) = 5.91; *p* < 0.05] (Appendix A). Similarly, diastolic BP was also affected by the opioid system blockade (CTRL: 77.67 mmHg ± 4.66 mmHg; NAL: 96.21 mmHg ± 14.34 mmHg; NAL-M: 97.18 mmHg ± 10.04 mmHg) [F(2, 12) = 5.53; *p* < 0.05] (Appendix A).

### 2.2. Effect of Selective Opioid Receptor Antagonists on Systolic and Diastolic Blood Pressure

Selective antagonism of each opioid receptor subtype was achieved via cyprodime hydrochloride (CYP), naltrindole hydrochloride (NTI) and nor-binaltorphimine dihydrochloride (nor-BNI), respectively for µ, δ and κ receptors. Systolic BP and diastolic BP were measured over 10 consecutive days (Figure 2). Systolic BP remained stable across the whole experiment as shown by non-significant time factor [F(3.42, 34.19) = 1.13; *p* = 0.36—CYP; F(2.39, 19.12) = 1.54; *p* = 0.24—NTI and F(3.54, 31.84) = 1.21; *p* = 0.32—nor-BNI]. Selective antagonism of µ, and κ opioid receptors caused an increase in systolic BP values [F(1, 10) = 25.38; *p* < 0.001—CYP; F(1, 9) = 38.54; *p* < 0.001—nor-BNI]. As observed, the blockade of δ opioid receptors altered systolic BP but did not achieve statistical significance [F(1, 8) = 4.13; *p* = 0.08—NTI]. The effect of pharmacotherapy was not time-dependent, as indicated by non-significant time × treatment interaction [F(9, 90) = 1.54; *p* = 0.15—CYP, F(9, 72) = 0.69; *p* = 0.71—NTI and F(9, 81) = 1.49; *p* = 0.17—nor-BNI].

Similarly, diastolic BP was altered by administration of selective µ [F(1, 10) = 22.77; *p* < 0.001—CYP] and κ [F(1, 9) = 38.00; *p* < 0.001—nor-BNI] antagonists but not by the blockade of δ opioid receptors [F(1, 8) = 3.26; *p* = 0.11—NTI]. Diastolic BP was stable across the duration of the experiment as indicated by non-significant time factor [*p* = 0.34—CYP; F(2.144, 17.15) = 0.98; *p* = 0.40—NTI; F(3.40, 30.62) = 1.42; *p* = 0.25—nor-BNI]. Moreover, antagonism of particular opioid receptors was not time-dependent, as revealed by non-significant time × treatment interaction [F(9, 90) = 1.18; *p* = 0.32—CYP; F(9, 72) = 1.53; *p* = 0.15—NTI and F(9,81) = 1.47; *p* = 0.17—nor-BNI].

Additionally, mean systolic BP and diastolic BP for the whole period of treatment were calculated for each individual. In a further step, those values were compared by one-way ANOVA (treatment as a factor) (Appendix A). Mean systolic BP calculated for whole period of treatment was affected after selective opioid system antagonism (CTRL: 102.72 mmHg ± 2.39 mmHg; CYP: 122.68 mmHg ± 2.86 mmHg; NTI: 115.134 mmHg ± 5.62 mmHg; nor-BNI: 117.77 mmHg ± 1.00 mmHg), as shown by significant treatment factor [F(3, 19) = 6.84; *p* < 0.01] (Appendix A). Similarly, diastolic BP was also elevated by the selective opioid system blockade (CTRL: 77.67 mmHg ± 4.66 mmHg; CYP: 97.73 mmHg ± 8.46 mmHg; NTI: 88.30 mmHg ± 12.31 mmHg; nor-BNI: 92.60 mmHg ± 3.39 mmHg) [F (3, 19) = 6.63; *p* < 0.01] (Appendix A).

### 2.3. Effect of Non-Selective Blockade of the Opioid System on the T-Cells Subpopulations

Subpopulations of splenic T-cells after 12 days of the opioid system antagonism were characterized by flow cytometry (Figure 3, Figure 4 and Figure 5). One-way ANOVA (treatment as an independent factor) revealed that the non-selective blockade of the opioid system affected a percentage of T-cells (CTRL: 35.81% ± 1.72%; NAL: 36.32% ± 2.38%; NAL-M: 24.09% ± 1.43%) [F(2, 13) = 11.93; *p* < 0.01] and CD8 T-cells (CTRL: 11.12% ± 1.78%; NAL: 10.20% ± 1.16%; NAL-M: 17.02% ± 1.17%) [F(2, 13) = 9.71; *p* < 0.01] but failed to induce changes in CD4 T-cells (CTRL: 72.17% ± 4.76%; NAL: 73.82% ± 4.36%; NAL-M: 70.68% ± 2.15%) [F(2, 13) = 0.16; *p* = 0.86] (Figure 3).

Furthermore, one-way ANOVA revealed that non-selective antagonism of the opioid system did not affect the percentage of activated CD8 T-cells (CTRL: 6.30% ± 0.97%; NAL: 9.71% ± 1.36%; NAL-M: 9.10% ± 0.92%) [F(2, 13) = 2.48; *p* = 0.12] nor CD4 T-cells (CTRL: 5.93% ± 0.48%; NAL: 7.74% ± 0.52%; NAL-M: 7.27% ± 0.52%) [F(2, 13) = 1.58; *p* = 0.24] (Figure 4).

Administration of NAL or NAL-M did not affect subpopulations of effector (CTRL: 7.30% ± 0.62%; NAL: 8.67% ± 1.80%; NAL-M: 8.10% ± 1.87%) [F(2, 13) = 0.19; *p* = 0.83], naive (CTRL: 64.84% ± 6.31%; NAL: 61.36% ± 4.13%; NAL-M: 57.76% ± 4.39%) [F(2, 13) = 0.48; *p* = 0.63] and central memory (CM) (CTRL: 18.69% ± 4.36%; NAL: 24.13% ± 3.03%; NAL-M: 31.16% ± 6.25%) [F(2, 13) = 1.77; *p* = 0.21] CD8 T-cells as shown by one-way ANOVA. However, we observed that non-selective antagonism of the opioid system caused an increase in the percentage of effector CD4 T-cells (CTRL: 18.41% ± 1.82%; NAL: 17.58% ± 2.06%; NAL-M: 25.03% ± 1.34%) [F(2, 13) = 5.16; *p* < 0.05] but did not affect naive (CTRL: 67.14% ± 3.68%; NAL: 69.32% ± 2.96%; NAL-M: 59.44% ± 1.81%) [F(2, 13) = 3.09; *p* = 0.08] and CM (CTRL: 7.96% ± 1.03%; NAL: 8.85% ± 0.65%; NAL-M: 11.49% ± 1.48%) [F(2, 13) = 2.84; *p* = 0.10] subpopulations of CD4 T-cells (Figure 5).

### 2.4. Effect of the Selective Blockade of the Opioid System on the T-Cell Subpopulation

Subpopulations of T-cells isolated from the spleen after 12 days of selective opioid receptor antagonism were characterized by flow cytometry (Figure 6, Figure 7 and Figure 8). One-way ANOVA (treatment as an independent factor) revealed that selective blockade of the opioid receptors did not affect the percentage of T-cells (CTRL: 35.81% ± 1.72%; CYP: 28.09% ± 2.53%; NTI: 33.05% ± 2.37%; nor-BNI: 31.66% ± 2.95%) [F(3, 19) = 1.37; *p* = 0.28], CD8 T-cells (CTRL: 11.12% ± 1.18%; CYP: 14.93% ± 3.78%; NTI: 15.51% ± 0.79%; nor-BNI: 15.84% ± 1.83%) [F(3, 19) = 1.03; *p* = 0.40] and CD4 T-cells (CTRL: 72.17% ± 4.76%; CYP: 75.83% ± 3.75%; NTI: 72.82% ± 1.90%; nor-BNI: 72.66% ± 2.50%) [F(3, 19) = 0.25; *p* = 0.86] (Figure 6).

Furthermore, one-way ANOVA revealed that selective antagonism of the opioid receptors affected the percentage of activated CD4 T-cells (CTRL: 5.93% ± 0.48%; CYP: 7.96% ± 0.50%; NTI: 8.08% ± 0.51%; nor-BNI: 7.09% ± 0.49%) [F(3, 19) = 3.53; *p* < 0.05]. Higher activation of CD8 cells was observed, but it failed to achieve statistical significance (CTRL: 6.30% ± 0.97%; CYP: 9.39% ± 1.07%; NTI: 8.81% ± 0.91%; nor-BNI: 7.21% ± 0.56%) [F(3, 19) = 2.55; *p* = 0.09] (Figure 7).

Administration of selective opioid receptor antagonists caused alterations in effector (CTRL: 7.30% ± 0.62%; CYP: 12.79% ± 1.26%; NTI: 8.92% ± 0.71%; nor-BNI: 8.34% ± 1.20%) [F(3, 19) = 4.87; *p* < 0.05] and CM (CTRL: 18.69% ± 4.36%; CYP: 29.68% ± 1.49%; NTI: 35.18% ± 3.81%; nor-BNI: 26.74% ± 3.15%) [F(3, 19) = 3.80; *p* < 0.05] subpopulations of CD8 T-cells as shown by one-way ANOVA. Additionally, we observed a tendency for a decreased percentage of naive CD8 T-cells; however, one-way ANOVA did not reveal statistical significance (CTRL: 64.84% ± 6.31%; CYP: 49.14% ± 0.78%; NTI: 51.63% ± 4.54%; nor-BNI: 58.75% ± 3.70%) [F(3, 19) = 2.51; *p* = 0.09]. Furthermore, we observed that selective antagonism of the opioid receptors induced changes in the percentage of naive (CTRL: 67.14% ± 3.68%; CYP: 56.75% ± 2.94%; NTI: 60.16% ± 1.27%; nor-BNI: 54.87% ± 3.24%) [F(3, 19) = 3.31; *p* < 0.05] and CM (CTRL: 7.96 ± 1.03; CYP: 8.85 ± 0.83; NTI: 12.30 ± 0.56; nor-BNI: 10.43 ± 0.91) [F(3, 19) = 4.80; *p* < 0.05] CD4 T-cells. A trend was also observed for an increased percentage of effector CD4 T-cells without reaching pre-defined statistical significance (CTRL: 18.41% ± 1.82%; CYP: 25.48% ± 1.70%; NTI: 23.56% ± 1.37%; nor-BNI: 28.83% ± 3.28%) [F(3, 19) = 2.71; *p* = 0.07] (Figure 8).

## 3. Discussion

Our results demonstrate that blockade of opioid receptors increased BP levels in mice selected for high swim-stress-induced analgesia. We have shown that this effect can also be achieved by blocking only peripheral opioid receptors using NAL-M, which does not cross the BBB [14].

Studies on humans have not been consistent with the demonstration of the effects of NAL on BP. Some report elevated BP levels upon NAL administration [15,16] whereas others fail to show any effect [17,18,19,20]. Similarly, investigations into the effect of naltrexone on human BP have had varied findings, either increasing [21,22] or neutral [23,24]. However, the closest animal model to humans (using rhesus monkeys) showed that opioid receptor blockade by NAL or naltrexone increased BP after morphine pretreatment [25]. A similar effect was achieved in squirrel monkeys using either opioid antagonist [26].

Inconsistent results may potentially be due to differences related to the level of endogenous opioid system activity of studied subjects, including the existence of possible polymorphisms among opioid receptors in the human population. Moreover, different doses of the tested compounds and regimens and a relatively small number of subjects involved, particularly in clinical trials, could be responsible for these discrepancies. Furthermore, it may suggest that the endogenous opioid system is active in hypotensive states but exerts little control over BP during states of pressor response, for example, during exercise.

In animal models, there is no consistency in results related to the role of opioid receptor antagonism on BP levels. Moreover, most of the experiments were performed in models comprising various diseases or pathologies. In our previous work, it has been described that mice with low swim-stress-induced analgesia had higher BP basal levels than HA mice. No changes in BP level were observed after NAL administration. By contrast, HA mice had elevated BP after NAL administration [13].

In this study, NAL or NAL-M was continuously released for 12 days via an osmotic minipump. An increase in systolic and diastolic BP in HA mice was observed. Opioid receptors couple predominantly to G proteins; upon binding, that leads to the inhibition of cAMP-mediated pathways, resulting in a decrease in BP [27]. By blocking opioid receptors with a potent antagonist, the opioid receptors can no longer modulate endogenous processes, including BP control. In our previous work, we have presented the important role of NAL in the regulation of guanylyl cyclase expression, resulting in BP changes [13]. A vital component of BP regulation is localized in the locus coeruleus of the brainstem, which has binding sites for opioids, noradrenaline, angiotensin II, etc. [28]. All of them are implicated in blood-pressure homeostasis in healthy conditions as well as under stress. Normally, opioid receptors of the locus coeruleus are predominantly µ opioid receptors. Endogenous β-endorphins bind to them, contribute to BP decrease, or display antihypertensive activity, which was demonstrated by intravenous injection of this peptide in several animal studies, particularly in chronic regimens [29,30]. Importantly, in the work presented here, we provide new data underlying the importance of peripheral opioid receptors in the regulation of BP. It demonstrates that the blockade of opioid receptors by both NAL (permeable to BBB) and NAL-M (impermeable to BBB) cause similar increases in BP. 

Opioid receptors are widely distributed in the periphery, including in the circulatory system, peripheral sympathetic nerves, and adrenal glands [31]. Opioid receptors are expressed by cardiomyocytes of the heart as well as vascular smooth muscle cells. They play modulatory functions, counteracting cardiac adrenergic stimulation via cAMP inhibition, thus lowering BP, providing cardioprotection, and influencing angiogenesis [32,33,34,35]. In addition, both cardiomyocytes and endothelial cells or smooth muscle cells are able to produce opioid peptides to self-regulate contractions of the heart and vessels [36]. Consistent results on BP upon chronic blockage of the opioid system were achieved in the study on dogs treated with NAL for 7 days (daily dose was 0.5 mg/kg/day) [37]. Mean arterial pressure was significantly elevated during the period of antagonist infusion (averaged MAP increase by 11 ± 1%) and decreased to baseline level after treatment completion. In the work demonstrated by Szilagyi et al., BP increase was associated with angiotensin I decrease in blood plasma, probably as a compensatory response to the rise in BP [37].

In young normotensive Wistar Kyoto (WKY) rats and spontaneously hypertensive rats (SHR) exposed to NAL prenatally, increased systolic BP was observed. The effect was maintained for 3 weeks postnatally in SHR rats and almost 2 weeks in normotensive animals and could not be restored via continuous treatment for an additional 2 weeks [38]. The animals were treated longer than in the work demonstrated here and with the period of uncontrolled concentration of drug administration in pups (receiving NAL in mother’s milk). These factors could influence the prolonged NAL-induced effect on systolic BP. Chronic treatment with NAL or naltrexone leads to an increase in opioid receptor density in the brain, as well as endogenous ligands that enhance agonist binding if subsequently administered [39,40]. Both in the work displayed here and the cited study, the disappearance of the NAL effect on BP after a fixed period could be observed, along with other compensatory mechanisms involving other GPCR receptors and developed tolerance to the drug. In addition, the NAL dose-dependent response curve has been demonstrated to resemble a U-shape or inverted U-shape [41]. Therefore, low doses are likely to be more effective (in BP increase and the effect maintenance) than higher concentrations.

Clonidine-induced reduction in BP and heart rate has been shown to be reversed by 2 mg NAL in SHR rats, though NAL alone at the same dose did not affect either BP or heart rate [42]. Clonidine binds to G_i_-coupled α2 adrenergic receptors and decreases noradrenaline release from central and peripheral sympathetic nerve endings, thus contributing to a decrease in BP. The use of NAL eliminates the opioid system from the modulation of BP and reverses clonidine action, possibly in response to clonidine-induced changes in catecholamine concentrations in plasma. It has been shown that clonidine reduces plasma levels of adrenaline and noradrenaline in SHR rats [43]. NAL increases plasma adrenaline, but not noradrenaline, in exercise conditions in humans. Adrenaline is released in excess but displays no effect on resting concentrations. Therefore, it probably reveals its effect in the state of altered catecholamine concentrations in plasma [44]. Animals were placed in a restrainer during a 90-minute experiment involving SHR rats [42]. This alone could influence the NAL effect when no other factor, physical or chemical, affects catecholamine levels. In our experiment, mice were allowed to move freely during NAL treatment, except for a short time for BP measurements. Additionally, a relatively low dose of an opioid antagonist, less than half of the dose of clonidine, could have importance on tissue distribution of catecholamines, if given alone or in combination [42].

The stimulating effect of NAL on BP has been documented by several studies on cats and dogs. In both, NAL administration in anesthetized animals leads to the elevation of BP [6,7]. As mentioned earlier, an increase in BP may be explained by NAL- or naltrexone-induced changes in the hormone levels, as vasopressin [45] and catecholamines [46] are implicated in BP modulation. Contrary to our results, in one study of SHR with experimental inflammation caused by a bacterial antigens cocktail, NAL (0.1 mg/kg) but not NAL-M (0.12 mg/kg) reduced mean arterial BP. Both NAL and NAL-M delayed the occurrence of hypertension in SHR in these chronic pain and inflammation conditions [47]. In another study on hypertensive patients with acute stress-induced increase in BP, NAL paradoxically decreased BP, which correlated with a decrease in plasma norepinephrine concentration. In the control group, NAL failed to modify any of the parameters [31].

In the work presented here, the aim was to elucidate the role of each opioid receptor in the regulation of BP. We found that antagonism of μ and κ but not δ led to the elevation of both systolic and diastolic BP. The effect of μ opioid receptor agonism on BP has been extensively described. Morphine and its derivatives cause a decrease in BP, and in line with that, antagonism of μ opioid receptor causes elevation of BP. However, the aim of this study was to investigate the role of endogenous opioids, as such conditions with exogenous opioid pretreatment were not used, as most other studies have. In the study by Chen et al., the reduction of mean arterial pressure caused by peripherally acting, μ opioid receptor agonist loperamide was abolished by the selective μ opioid receptor antagonist CYP in spontaneously hypertensive rats (SHRs) but not in the normal group of WKY rats [34]. In our study, μ opioid receptor inhibition using CYP resulted in a BP decrease across almost the entire treatment period. The reasons for the different actions of CYP in the work by Chen et al. remain speculative since there are no other reports on this μ receptor antagonist effect on BP. The distribution of opioid receptors differs between SHR rats and control strains, both in central brain regions and peripherally in the heart [33,48].

Selective κ opioid receptor antagonist (nor-BNI) caused elevation of BP in our experiment; however, the effect was revealed with a one-day delay, and significantly increased BP could only be recorded on the last day of the experiment (day 12). This compares to the CYP group with the last observed significant effect on day 11, while the response curve displayed similar fluctuations in BP values. It has been documented that the pharmacodynamics of nor-BNI in vivo differ greatly from other opioid antagonists [49]. The achievement of maximal κ opioid receptor antagonism may be delayed by hours or days compared to minutes for NAL. Furthermore, the duration of the nor-BNI-mediated effect is very long. Typical competitive opioid antagonists are effective for hours or, at most, days. Similarly, κ antagonism can be maintained for weeks or even months. Initially, it was considered that nor-BNI metabolism is slow, but it turned out this compound can activate c-Jun N-terminal kinase 1 (JNK1), causing desensitization of κ opioid receptors that may last long after this antagonist elimination. In addition, nor-BNI has been shown to display binding affinity to the α2_C_-adrenergic receptor (K_i_ = 630 nM), accounting for some difference in the response pattern of this compound seen in our study [49].

Consistently, nor-BNI (4 nmol/kg/h) administered unilaterally into the hippocampus for 14 days increased the BP of SHR rats from baseline 136 ± 11 to 150 ± 10 mmHg with a significant peak on day 7 of the treatment. This was alongside the complete absence of any effect of nor-BNI on BP in control WKY rats [50]. Importantly, more effective blockage of κ opioid receptors resulted in an apparent increase in BP both in SHR and normotensive rats has been achieved after bi-hippocampal microinjection of antisense oligodeoxynucleotide to the rat κ opioid receptor twice a day for 5 days [51]. Certainly, bilateral treatment of the specific brain area inducing hypertension in both rat strains appeared to be more effective than the unilateral approach [50]. The same BP-increasing effect was observed when nor-BNI (10 nmol/0.5 mL twice a day for 13 days) was administrated bilaterally to the hippocampus in both normotensive rats and animals with social isolation-induced hypertension [52]. These data suggest that the rat hippocampal κ opioid receptor and the endogenous neuropeptide dynorphin may be involved in the central neural regulation of BP [52]. In the swine model, nor-BNI produced a modest increase in arterial pressure and heart rate after five days of administration [53]. However, in another study, the central administration of nor-BNI was unable to modify BP in sodium-depleted Wistar rats [54].

Blockade of δ opioid receptors with NTI in mice had an insignificant effect on BP in our experiment set. In our work we did not observe an effect of NTI on BP level; however, another group has demonstrated that in young hypertensive SHR rats, four weeks of antagonism of δ opioid receptors by ICI 154 129 compound (10 µg/h sc.) caused a lowering of BP seen after 3 weeks of treatment (with measurements performed twice a week) [55]. In these chronically treated rats, an increase in β-endorphins but not enkephalins has been detected in the hypothalamus, which reflects the initiation of compensatory processes, counteracting the persistent inhibition of δ opioid receptors by the reprogramming of opioid peptide synthesis towards the enhanced production of endogenous ligands for unoccupied µ opioid receptors [55]. Since µ opioid receptor agonism usually contributes to hypotension, the effect authors observed likely could be assigned to β-endorphin action. Another study reported that BP and heart rate were both increased in cocaine-treated rats by central administration of 100 μg of NTI that blocked the bradycardiac effect of cocaine [56]. However, in conscious rats, NTI (1 mg/kg), when injected intravenously, failed to change any systemic cardiovascular parameters [57]. Similarly, results from a further study performed on rabbits are in line with our data, showing no effect of δ opioid receptor antagonism on BP level (monitored for 30 min) when NTI was injected into the central nervous system [58]. The researchers used L-glutamate microinjection to localize the rostral ventrolateral medulla (RVLM) pressor area first and then administered the antagonist of δ opioid receptors [58]. All these seemingly contradictory influences of NTI on BP reported are, in fact, results of various factors (such as pretreatment with distinct pharmacological agents) and experimental conditions.

Besides the direct effect of opioid system on BP level, opioids also modulate immune system function found recently to have an important role in hypertension development. The important study performed by Guzik et al. provided data for the substantial role of T-cells and their activation in hypertension development [59]. Exogenous opioids such as morphine and fentanyl have been found to impair the function of macrophages, natural killer cells, and T-cells. Prolonged morphine treatment weakens the adaptive immune response; for example, it impairs T-cell function, alters the expression of cytokines, suppresses T-cell apoptosis, and modifies T-cell differentiation [60]. Taking into account the above-mentioned findings, we attempted to determine whether modulation of T-cells by opioid receptor antagonists may play a role in the regulation of BP in mice selected for high analgesia induced by swim stress.

Our results clearly show that antagonism of peripheral opioid receptors decreased the percentage of T-cells but increased the percentage of cytotoxic CD8 subpopulation within T-cells in murine spleens. We did not observe these changes when NAL was used. Treatment of humans and rats with NAL or naltrexone induces NAL-enhanced proliferation of isolated, mitogen-stimulated T-lymphocytes upon acute intravenous administration of NAL and inhibited by naltrexone after eight days of treatment [61]. This is equally indirect evidence of the immunosuppressive effect of opioids on immune cells. Simultaneously, splenocytes derived from chronically NAL-treated rats used in β-endorphin binding studies demonstrated compensatory upregulation of opioid receptors [61]. Interestingly, 2-hour swimming-induced stress in rats caused a reduction of splenic T-lymphocyte proliferation, and treatment with naltrexone inverted this effect. However, only a single session was able to induce that effect [62]. This is likely due to the acute release of immunosuppressive opioids upon extensive physical exercise and stress in the first session and subsequent desensitization of continuous, endogenous ligands-stimulated opioid receptors upon repeated stress conditions (the model used was 2 hours swim/day for 5 days). NAL-reversible T-cell decrease demonstrated an opioid receptor-mediated effect that the authors observed. On the other hand, another study showed that methionine–enkephalin analogs enhanced human T-cell proliferation in vitro when incubated for 5 days. Their immunomodulatory activities were completely blocked by NAL. Prolonged treatment with opioid agonists led to opioid receptor desensitization that switched off the inhibitory effect of opioids along with the upregulation of compensatory endogenous mitogen-mediated pathways. The treatment of the cultured human T-lymphocytes with selective opioid receptor antagonists revealed that observed immunomodulatory properties are mediated mainly through the δ opioid receptor, and partially via μ opioid receptors, but not by κ opioid receptors [63]. Similar effects were reported by Jiao et al. Selective blocking of μ and δ opioid receptors resulted in inhibition of the CD8 T-cell proliferation and the expressions of surface molecules, while treatment with methionine–enkephalin promoted the expression of these molecules including FasL, which might be a predictor of apoptosis upon prolonged treatment [64]. This is in line with our data showing that T-cells predominantly express δ opioid receptors on their surface. Expression of early activation markers on T-cells was measured in our study. We have observed an increase in CD69 expression after NAL treatment; however, it failed to reach statistical significance (*p* = 0.053). Moreover, NAL-M treatment, but not NAL treatment, caused an elevation of the effector CD4 T-cell population. Other studies have shown that low-dose naltrexone can increase the expression of MHC class II molecules on antigen-presenting cells (APCs), such as dendritic cells and macrophages [65]. MHC class II molecules are essential for presenting antigens to helper T-cells (Th), which provide specific help for B-cells and cytotoxic T-cells [66]. NAL can also increase the production of IL-12, another cytokine that promotes helper T-cells differentiation [65].

Immunosuppressive effects caused by opioid receptor agonists are mediated mainly through μ and δ opioid receptors; however, some studies also include κ opioid receptor agonists as a potent immunosuppressant [67,68]. We found that the blockade of μ and δ opioid receptors is associated with increased expression of T-cell activation marker CD69 on CD4 T-cells. T-cell activation via δ opioid receptor agonism may be due to calcium mobilization. The agonism of δ opioid receptors elevates calcium mobilization, which modulates downstream pathways responsible for the modulation and function of T-cells. Antagonism of the δ opioid receptor reversed this effect [69,70]. Interestingly, in our study, we observed an increase in central memory CD4 and central memory CD8 T-cell populations after NTI treatment. In this process, the μ opioid receptor also plays a role since, after CYP administration, an increase of CD8 effector and central memory T-cell populations was observed. Suppression of immune function by δ opioid receptor antagonists itself was described as well. In vitro exposure to δ opioid receptor antagonist resulted in an apparent dose-related suppression of B-cell proliferation, cytokine production by T-helper cells, and natural killer (NK) cell activity [71,72]. Similarly, only δ opioid receptors were found to be expressed on CD4 T-cells, but only after stimulation with concanavalin A [73]. Expression of μ opioid receptors was only detected on T-cells and B-cells when stimulated by IL-4 or TNF-alpha. This indicates that μ opioid receptor expression may be altered under conditions such as inflammation, infection, or neurological conditions associated with imbalanced cytokine expression [74]. Morphine-treated peripheral blood mononuclear cells decreased IL-2 and IFN-gamma and increased IL-4 and IL-5 production. Changes in cytokine synthesis were abolished in μ opioid receptor knockout mice [75].

Another opioid receptor antagonist we studied was nor-BNI—a κ opioid receptor antagonist. In our study, we have not detected κ opioid receptors on T-cells (at an mRNA expression level). However, on Jurkat T-cells, the expression of κ opioid receptors was reported. Furthermore, the administration of μ, δ, and κ opioid agonists induced a chemotactic response in Jurkat cells [76]. In our study, nor-BNI decreased the percentage of naive CD4+, but that effect could be opioid receptor independent since this κ opioid antagonist is able to cross-talk to other receptor types [49].

## 4. Methods

### 4.1. Animals

In the present study, we used 12-week-old male mice from a selectively bred, HA mouse line originating from outbred Swiss Webster mice. A previously described protocol was used to maintain the phenotypic traits of HA mouse lines [77]. Briefly, mice of either sex were subjected to 3 minutes of forced swimming in 20 °C water; after that, individual mice were placed in clean shoebox cages for a 2-minute rest period. Each mouse was then screened for the latency of a nociceptive reflex in a hot-plate test. Those mice that displayed the longest latency time (50–60 s) of the hind paw or lick response (whichever occurred first) were selected as progenitors of the HA mouse line. A similar procedure was repeated in each offspring generation, but only subjects demonstrating the longest post-swim hot-plate latencies were mated to maintain the lines. Mice were kept in standard cages (3–4 individuals of the same sex) in a controlled environment: 22 ± 2 °C temperature, 55 ± 5% humidity, with a 12 h–12 h light–dark cycle (lights on at 7 am). Naturally, all individuals had ad libitum access to standard chow (LABOFEED H, Kcynia, Poland) and tap water. Experiments on live mice were conducted according to the ethical approval (decision no. WAW2/180/2019 and WAW2/058/2020) received from the II Local Ethics Committee for Experiments on Animals in Warsaw.

### 4.2. Antagonist Administration

To study the overall effect of the opioid system on BP level, non-selective opioid system antagonists were used: naloxone hydrochloride (NAL) or naloxone methiodide (NAL-M), which does not cross the BBB [14]. Selective antagonism of each opioid receptor subtype was achieved via cyprodime hydrochloride (CYP), naltrindole hydrochloride (NTI), and nor-binaltorphimine dihydrochloride (nor-BNI), respectively, for µ, δ, and κ opioid receptors. All pharmacological agents were dissolved in vehicle solution (0.9% NaCl) and administered via an implanted osmotic minipump (Alzet Model 2002; Alzet, Cupertino, CA, USA) for 12 days at a dose of 5 mg/kg/day. Briefly, anesthetized mice had an incision made on the back, a pocket was formed, a minipump was inserted, and the wound was closed. In sham procedure vehicle solution was administered via minipump (control group—CTRL). Each experimental group included 5–7 animals. NAL and CYP were purchased from Tocris Bioscience (Bristol, UK); NAL-M, NTI, and nor-BNI were purchased from Merck (Darmstadt, Germany).

### 4.3. Measurement of Blood Pressure

BP was measured before and during the infusion of the appropriate drug using a non-invasive tail–cuff measurement system (Coda System, Kent Scientific, Torrington, CT, USA). Before minipump implantations, to minimize the stress effect occurring during measurements, mice were familiarized with tail–cuff BP measurements for one week. After minipump implantation, mice were rested for 2 days for post-procedure recovery. BP was recorded for the next 10 days. Each session of BP measurement included 5 preliminary measurements and 10 actual measurements, which were analyzed. Preliminary measurements were performed to allow the animals to warm up sufficiently to obtain good blood flow in the tail.

### 4.4. Analysis of T-Cell Subpopulation and Activation

Spleens were harvested and mashed through 70-μm strainers (VWR International, Avantor, Radnor Township, PA, USA) to isolate single cells. RBC lysis buffer (Biolegend, San Diego, CA, USA) was used to deplete red blood cells. Splenocytes were stained in FACS buffer for 20 min at 4 °C in the dark with the monoclonal antibodies [Appendix A]. Cells were analyzed by a CytoFLEX flow cytometer (Beckman Coulter, Brea, CA, USA), and data were analyzed using Flow Jo v.10 (Ashland, OR, USA). For each experiment, fluorescence-minus-one controls (FMO) were performed. In selected experiments, FMO gating strategies were confirmed by isotype controls.

### 4.5. Data and Statistical Analysis

Gathered data were tested for normality by the Shapiro–Wilk test and scanned for possible outliers with the ROUT method (Q = 1%). For comparison of BP measurements, two-way RM-ANOVA with Geisser–Greenhouse correction was used (treatment and time as independent factors). To assess the effect of treatment in flow cytometry analysis, one-way ANOVA (treatment as an independent factor) was used. In the case of significant differences indicated by ANOVA, multiple comparisons were calculated with the use of the Fisher LSD test. Differences were considered to be significant when the *p*-value was less than 0.05. GraphPad Prism was used to calculate all analyses and visualize data. All data presented in the figures are mean ± SEM.

## 5. Conclusions

In conclusion, our results provide new insight into the mechanisms of BP regulation via the endogenous opioid system. Using a unique murine model with a hyperactive opioid system, we can evaluate this effect without using exogenous opioids to stimulate opioid receptors. Our results indicate κ and μ opioid receptors are mostly involved in the regulation of BP under conditions of an enhanced endogenous opioid system. By contrast, the expression of opioid receptors on T-cells is marked by the dominant presence of δ opioid receptors and the relatively low or absent expression of other opioid receptors. These findings suggest that the mechanism of regulation of BP can operate independently of T-cell activation induced by opioid receptors. Therefore, more research is needed to understand how opioid receptor antagonists interact with other factors involved in immune regulation.

## 6. Limitations

One of the limitations of this work is the model we used. A mouse model of high swim stress-induced analgesia characterized by enhanced endogenous opioid system activity enabled us to record and magnify even small changes induced by the blockade of the opioid receptors. Despite its usefulness, this model is not widely utilized in the research community, which limits discussion and the reference of our results to similar studies. Additionally, due to the enhanced endogenous opioid system, the observed effect might be overestimated when translated to other mouse models or human studies. Furthermore, the determination of the endogenous opioid levels in the blood or other tissues could support the explanation of the observed effect of applied antagonists, including defining compensatory mechanisms. 

## Figures and Tables

**Figure 1 ijms-25-02618-f001:**
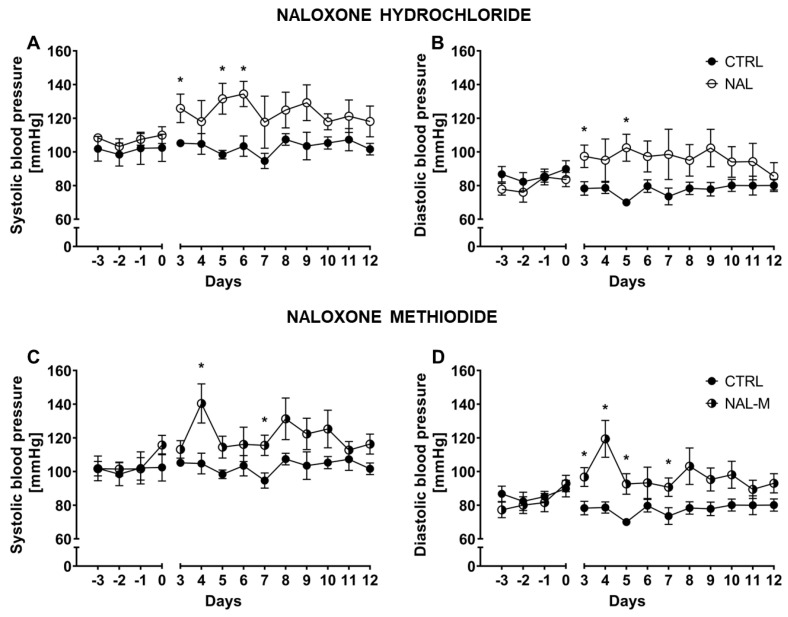
Effect of naloxone hydrochloride (**A**,**B**) and naloxone methiodide (**C**,**D**) on systolic and diastolic BP in HA mice (*n* = 5). Post hoc comparisons vs. control group are denoted by *. One symbol represents *p*-value < 0.05.

**Figure 2 ijms-25-02618-f002:**
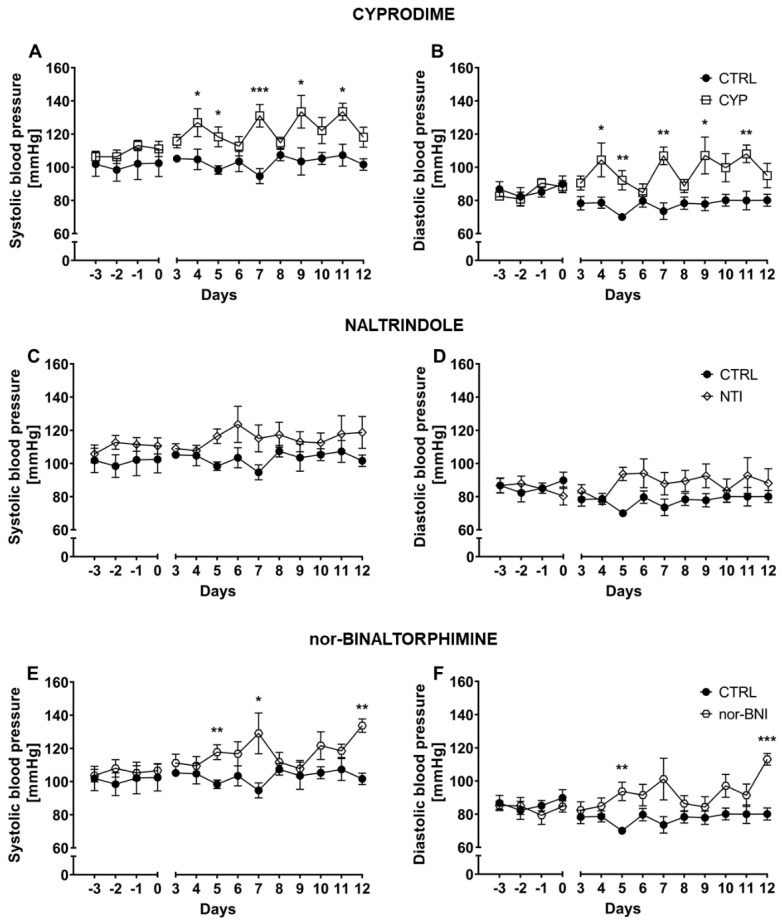
Effect of cyprodime (**A**,**B**), naltrindole (**C**,**D**), and nor-binaltorphimine (**E**,**F**) on systolic and diastolic BP in HA mice (*n* = 5–7). Post hoc comparisons of the treatment group vs. control group are denoted by *. One, two or three symbols represent *p*-value < 0.05, <0.01, <0.001, respectively.

**Figure 3 ijms-25-02618-f003:**
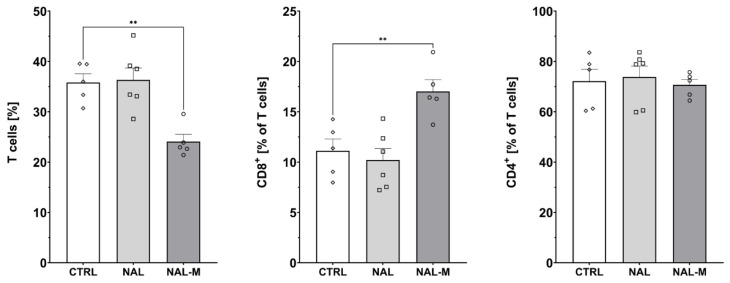
Percentage of T-cells and T-cells subpopulations isolated from spleen after non-selective blockage of the opioid system. Post hoc comparisons vs. control group are denoted by *. Two symbols represent *p*-value < 0.01. Abbreviations: CTRL—control group; NAL—naloxone; NAL-M—naloxone methiodide.

**Figure 4 ijms-25-02618-f004:**
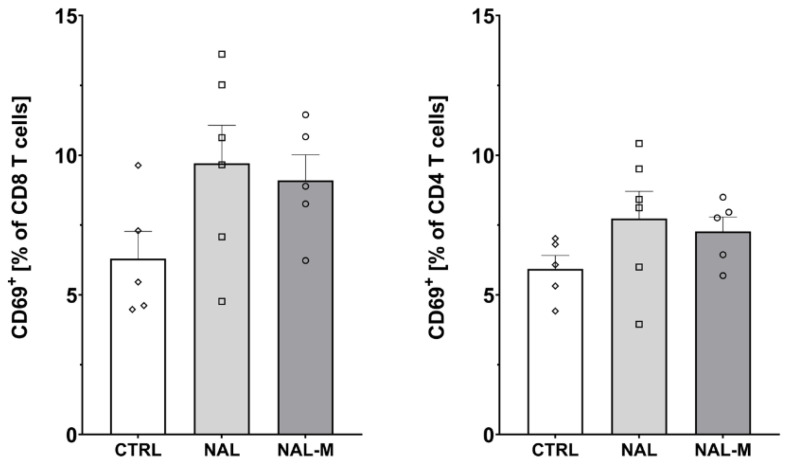
Percentage of activated CD8 and CD4 T-cells isolated from spleen after non-selective blockage of the opioid system. No statistically significant results were obtained. Abbreviations: CTRL—control group; NAL—naloxone; NAL-M—naloxone methiodide.

**Figure 5 ijms-25-02618-f005:**
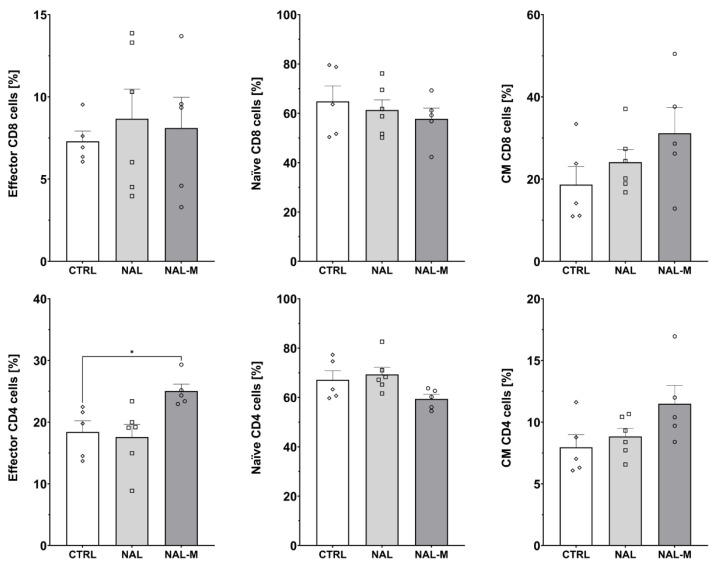
Percentage of effector, naive, and central memory (CM) cells among CD8 and CD4 T-cells subpopulations isolated from spleen after non-selective blockade of the opioid system. Post hoc comparisons vs. control group are denoted by *. One symbol represents *p*-value < 0.05. Abbreviations: CTRL—control group; NAL—naloxone; NAL-M—naloxone methiodide.

**Figure 6 ijms-25-02618-f006:**
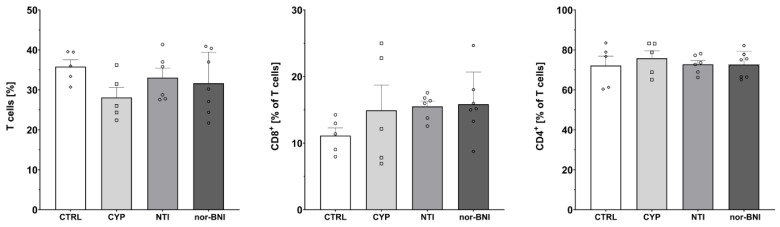
Percentage of T-cells and T-cell subpopulations in spleen after a selective blockade of the opioid system receptors. Abbreviations: CTRL—control group; CYP—cyprodime; NTI—naltrindole; nor-BNI—nor-binaltorphimine.

**Figure 7 ijms-25-02618-f007:**
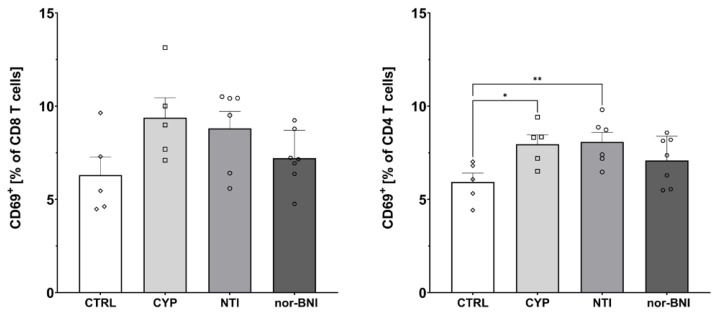
Percentage of activated CD8 and CD4 T-cells in spleen after selective blockade of the opioid system receptors. Post hoc comparisons vs. control group are denoted by *. One and two symbols represent *p*-values < 0.05 and <0.01, respectively. Abbreviations: CTRL—control group; CYP—cyprodime; NTI—naltrindole; nor-BNI—nor-binaltorphimine.

**Figure 8 ijms-25-02618-f008:**
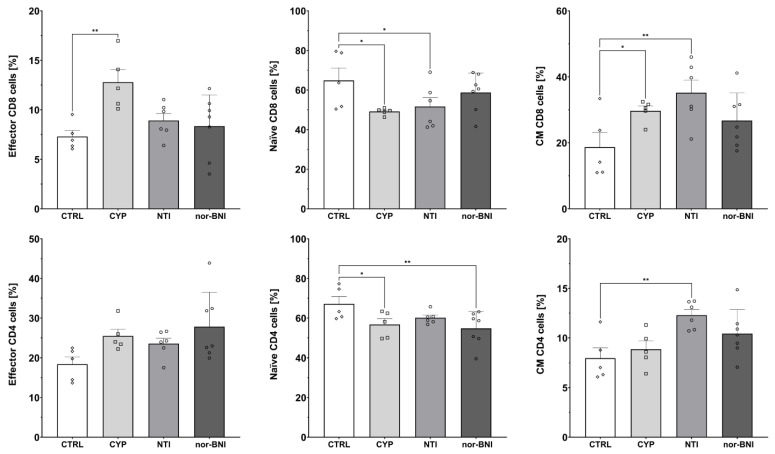
Percentage of effector, naive, and central memory (CM) cells among CD8 and CD4 T-cells subpopulations in spleen after selective blockade of the opioid system receptors. Post hoc comparisons vs. control group are denoted by *. One and two symbols represent *p*-values of <0.05 and <0.01, respectively. Abbreviations: CTRL—control group; CYP—cyprodime; NTI—naltrindole; nor-BNI—nor-binaltorphimine.

## Data Availability

The data presented in this study are available on request from the corresponding author.

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
