# Peer review of "The Role of Opioid Receptor Antagonists in Regulation of Blood Pressure and T-Cell Activation in Mice Selected for High Analgesia Induced by Swim Stress"

_ijms, 2024, doi:10.3390/ijms25052618_

Round 1

Reviewer 1 Report

Comments and Suggestions for Authors

The Manuscript ijms-2853416-peer-review-v1

Manuscript ijms-2853416-peer-review-v1 describes the effect of opioids receptor antagonists on blood pressure and immune system in mice model of high analgesia induced by swim test.

The manuscript is within the scope of International Journal of Molecular Sciences. Authors have successfully applied pharmacological tools to elucidate the effect of opioid receptor antagonists on blood pressure as well as on immune system. However, the following comments need to be considered.       

Major concerns:

1.       The abstract section is missing.

2.       Authors were unable to emphasise the rationale of conducting the study in the introduction section.  

3.       It seems that a key concept is missing in the current study. Authors need to clarify whether the development of mice model of high analgesia induced by swim test per se could alter blood pressure and immune cell proliferation. Then, pharmacological tools such as opioid receptor antagonists could be used to clarify these effects.

4.       The discussion section needs substantial editing. The current version of the discussion appears as an extensive literature review.

Comments on the Quality of English Language

1. The full term of abbreviation should be described in full as it first appears in the text. 

2. There are extra spaces in different parts of the manuscript. 

3. Authors should be consistent in term of using full terms or abbreviations.  

Author Response

Dear Reviewer,

Thank you for the effort put to review our paper and for valuable comments. We have now addressed all Reviewers’ comments and a point by point response is provided below. In addition, all changes are highlighted in the text of the revised manuscript.

“1.       The abstract section is missing.”

Thank you for pointing that out. The abstract has been added to the manuscript.

“2.       Authors were unable to emphasise the rationale of conducting the study in the introduction section.  “

The rationale to conduct this study was to evaluate which opioid receptors play the main role in regulation of blood pressure in a mouse model of high swim stress induced analgesia. It is important cause there is no paper showing effect of selective opioid receptor antagonists on BP neither T cells activation. Our previous experiments with naloxone showed that naloxone increased blood pressure level in HA mice but not LA mice. Moreover knowing that T cells activation is important in regulation of blood pressure in hypertension we have analyzed subsets of T cells and CD69 marker expression on T cells.

“3.       It seems that a key concept is missing in the current study. Authors need to clarify whether the development of mice model of high analgesia induced by swim test per se could alter blood pressure and immune cell proliferation. Then, pharmacological tools such as opioid receptor antagonists could be used to clarify these effects.”

Thank you for the comment. We have publish previously a paper (Hypertensive Effect of Downregulation of the Opioid System in Mouse Model of Different Activity of the Endogenous Opioid System - PMID: 33920718) where we analysed blood pressure changes in low analgesia (LA) and high analgesia (HA) mice. The current paper is continuation of our earlier research. We have added a few sentences of explanation to the present paper.

“4.       The discussion section needs substantial editing. The current version of the discussion appears as an extensive literature review.”

Thank you for the comment. The discussion section has been substantially changed.

“Comments on the Quality of English Language

  1. The full term of abbreviation should be described in full as it first appears in the text. “

Thank you for pointing this out. Once again manuscript have been carefully checked and abbreviation were described in full as it first appeared in the text.

“2. There are extra spaces in different parts of the manuscript. “

Thank you for pointing this out. Once again manuscript have been carefully reviewed and double-spaces were eliminated.

“3. Authors should be consistent in term of using full terms or abbreviations.  “

Thank you for pointing this out. We have made necessary corrections in the text.

Best regards,

Dominik Skiba (on behalf of all authors)

Reviewer 2 Report

Comments and Suggestions for Authors

The manuscript titled 'The role of opioid receptor antagonists in regulation of blood pressure and T cells activation in mice selected for high analgesia induced by swim stress' provides a comprehensive investigation into the effects of opioid receptor antagonists on blood pressure and T cell activation in a unique mouse model. The study employs selective opioid receptor antagonists to evaluate their impact on blood pressure and T cell activation in mice with enhanced endogenous opioid systems. The research findings shed light on the differential effects of various opioid receptor antagonists on blood pressure and T cell subpopulations, providing valuable insights into the complex interactions between opioid receptors, blood pressure regulation, and immune system function.

The manuscript presents a thorough review of the literature, providing a strong theoretical foundation for the study. The methods section is detailed and clearly outlines the experimental procedures, including the use of selective opioid receptor antagonists and the measurement of blood pressure and T cell subpopulations. The results are well-documented and demonstrate that blockade of opioid receptors, particularly μ and κ, leads to an increase in blood pressure levels, while antagonism of δ opioid receptors alters systolic blood pressure but does not reach statistical significance. Additionally, the study found that non-selective blockade of the opioid system reduces the percentage of T cells and increases the percentage of CD8 subpopulation within T cells in murine spleens.

The manuscript effectively discusses the implications of the findings, highlighting the potential role of opioid receptors in regulating blood pressure and T cell activation. The authors provide a critical analysis of the results, acknowledging the limitations of the study and the need for further research to fully understand the mechanisms underlying the effects of opioid receptor antagonists on blood pressure and immune system function.

Overall, the manuscript makes a significant contribution to the understanding of the opioid system's role in hypertension development and immune system modulation. The study's rigorous methodology and comprehensive analysis of the results strengthen the validity of the findings. However, further research is warranted to elucidate the specific mechanisms by which opioid receptor antagonists interact with other factors involved in immune regulation and blood pressure regulation.

Comments on the Quality of English Language

Minor editing of English language required.

Author Response

Dear Reviewer,

Thank you for the effort put to review our paper and for valuable comments. All changes are highlighted in the text of the revised manuscript. We agree that further research are necessary to elucidate the specific mechanisms by which opioid receptor antagonists interact with other factors involved in immune regulation and blood pressure regulation.

“Minor editing of English language required.”

The manuscript was carefully reviewed and English language was improved.

Best regards,

Dominik Skiba (on behalf of all authors)

Reviewer 3 Report

Comments and Suggestions for Authors

The study aimed to investigate the effect of opioid receptors on blood pressure (BP) by using selective opioid receptor antagonists and evaluate their effect on BP and T cells activation in this model. The current proposal is interesting and well-written. Therefore, I recommend that the current study be published after major revisions as follows:

1-   Please include the abstract part.

2-   The authors mentioned that ‘’ Our results indicate kappa and mu-opioid receptors 406 are mostly involved in regulation of blood pressure under conditions of enhanced endogenous opioid system’’ Please discuss the deeply possible mechanisms for these findings

3-   The authors also referred to ‘’. However, the expression of delta opioid receptors was most profound in T cells. These findings suggest that the mechanism of regulation of blood pressure and T cells activation by opioid receptors can operate independently of each other’’. How can opioid receptors operate independently of each other although both kappa and mu-opioid receptors are promising targets for T cells and macrophages?

4-   Could the authors highlight the role of hypertension and opioid receptors in COVID-19 infection?

References:

Hypertension is still a moving target in the context of COVID-19 and post-acute COVID-19 syndrome. J Med Virol. 2023 Jan;95(1):e28128. doi: 10.1002/jmv.28128.

Prevalence of COVID-19 outcomes in patients referred to opioid agonist treatment centers. Front Pharmacol. 2023 Mar 23;14:1105176. doi: 10.3389/fphar.2023.1105176.

5-    Please add a diagrammatic figure to propose the possible mechanistic pathway for these findings.

Author Response

Dear Reviewer,

Thank you for the effort put to review our paper and for valuable comments. We have now addressed all Reviewers’ comments and a point by point response is provided below. In addition, all changes are highlighted in the text of the revised manuscript.

“1-   Please include the abstract part.”

Thank you for pointing that out. The abstract has been added to the manuscript.

“2-   The authors mentioned that ‘’ Our results indicate kappa and mu-opioid receptors are mostly involved in regulation of blood pressure under conditions of enhanced endogenous opioid system’’ Please discuss the deeply possible mechanisms for these findings”

We have discussed that topic broader in the discussion section. All changes are highlighted in the text of the revised manuscript.

“3-   The authors also referred to ‘’. However, the expression of delta opioid receptors was most profound in T cells. These findings suggest that the mechanism of regulation of blood pressure and T cells activation by opioid receptors can operate independently of each other’’. How can opioid receptors operate independently of each other although both kappa and mu-opioid receptors are promising targets for T cells and macrophages?”

Thank you for your comment. We have changed these sentences to be more clear.

“4-   Could the authors highlight the role of hypertension and opioid receptors in COVID-19 infection?

References:

Hypertension is still a moving target in the context of COVID-19 and post-acute COVID-19 syndrome. J Med Virol. 2023 Jan;95(1):e28128. doi: 10.1002/jmv.28128.

Prevalence of COVID-19 outcomes in patients referred to opioid agonist treatment centers. Front Pharmacol. 2023 Mar 23;14:1105176. doi: 10.3389/fphar.2023.1105176.”

 Thank you very much for your comment. We agree that COVID-19 related papers are now in the spot. We carefully read mentioned articles. One of them described the role of hypertension in the context of COVID-19 and another one described lack of association between COVID-19 severity and methadone treatment. In our opinion it is out of scope of our article to focus on such a specific disease as COVID-19. If we would add this disease in the discussion section then we would need to discuss also bunch of other viral and bacterial infections associated with hypertension.

Despite of that we are grateful for that comment and we share your curiosity to link our findings with COVID-19.  We think that would be interesting topic for future studies.

“5-    Please add a diagrammatic figure to propose the possible mechanistic pathway for these findings.”

In our opinion there is still not enough data to construct a useful schematic pathway. There would be too many gaps and the full picture would be unclear. Further research are necessary to elucidate the specific mechanisms by which opioid receptor antagonists interact with other factors involved in immune regulation and blood pressure regulation.

Best regards,

Dominik Skiba (on behalf of all authors)

Reviewer 4 Report

Comments and Suggestions for Authors

In presented manuscript the Authors analyzed the role of different types of opioid receptors in regulation of blood pressure and T cells activation by using non-selective and MOR, DOR and KOR selective antagonists. The presented manuscript consider a very important problem which is the increasing incidence of hypertension and the information included in this article could be important in a process of drug development. Overall, I believe that it may be of reasonably high interest for the readers. However, after carefully reading this article, I conclude that although it is properly developed in accordance with the requirements for scientific publications, the Authors could not avoid certain shortcomings that do not allow the article to be published in its current form.

·       * The Authors must add the following sections: abstract, key words, conclusions, limitations. They are missing in the current version.

·      *  In my opinion, the Authors should add more information about used animal model. Additionally, it is worth expanding on the information included in lines 39-42.

·     *  The Authors should add a description of the minipump implantation in the methodology.

·       * There are no values for the ANOVA analyzes in the description of the results.

·      *  However, my biggest doubts are the actual benefits obtained from the presented experiments. Where do the Authors think novelty is located? Especially since the Authors in the discussion repeatedly emphasize that similar effects have been observed in other animal species and even in humans. Moreover, They emphasize that, for example, MOR antagonism on BP has been well described (line 305). This problem particularly concerns the effect of antagonists on BP and, to a much lesser extent, on T cell activation. In my opinion, this point requires a comment from the Authors.

Author Response

Dear Reviewer,

Thank you for the effort put to review our paper and for valuable comments. We have now addressed all Reviewers’ comments and a point by point response is provided below. In addition, all changes are highlighted in the text of the revised manuscript.

“The Authors must add the following sections: abstract, key words, conclusions, limitations. They are missing in the current version.”

Thank you for pointing that out. The abstract, key words, conclusions and limitations have been added to the manuscript.

“ In my opinion, the Authors should add more information about used animal model. Additionally, it is worth expanding on the information included in lines 39-42.”

Thank you for the comment. We have described that topic broader and provided more information about used animal model.

“The Authors should add a description of the minipump implantation in the methodology.”

Thank you for your comment. We have added short description of osmotic minipump implantation in the methodology section.

“There are no values for the ANOVA analyzes in the description of the results.”

Thank you very much for your comment. Values for ANOVA analyzes have been added to results section.

“However, my biggest doubts are the actual benefits obtained from the presented experiments. Where do the Authors think novelty is located? Especially since the Authors in the discussion repeatedly emphasize that similar effects have been observed in other animal species and even in humans. Moreover, They emphasize that, for example, MOR antagonism on BP has been well described (line 305). This problem particularly concerns the effect of antagonists on BP and, to a much lesser extent, on T cell activation. In my opinion, this point requires a comment from the Authors.”

Thank you for your comment. We have described that topic broader in the discussion section. All changes are highlighted in the text of the revised manuscript.

Best regards,

Dominik Skiba (on behalf of all authors)

Reviewer 5 Report

Comments and Suggestions for Authors

Skiba et al provided a new insight in mechanisms regulating blood pressure level by endogenous opioid system. By employment of unique model of mice with hyperactive opioid system, they could evaluate this effect without using exogenous opioids to stimulate opioid receptors. Their results indicate kappa and muopioid receptors are mostly involved in regulation of blood pressure under conditions of enhanced endogenous opioid system. In contrast, the expression of opioid receptors on T-cells is marked by dominant presence of delta opioid receptors and relatively low or absent expression of other opioid receptors. However, expression of delta opioid receptors was most profound on T cells. These findings suggest that mechanism of regulation of blood pressure and T cells activation by opioid receptors can operate independently of each other. The focus of the study and the results obtained are very interesting. I have a few questions as follows.

major concerns)

1) Please provide an abstract since you do not have one.

Author Response

Dear Reviewer,

Thank you for the effort put to review our paper and for valuable comments. All changes are highlighted in the text of the revised manuscript.

“1) Please provide an abstract since you do not have one.”

Now the abstract has been added to the manuscript.

Best regards,

Dominik Skiba (on behalf of all authors)

Round 2

Reviewer 1 Report

Comments and Suggestions for Authors

The authors have addressed the comments of interest. 

Reviewer 3 Report

Comments and Suggestions for Authors

The authors have successfully addressed the comments